# Exposure-Response and Clinical Outcome Modeling of Inhaled Budesonide/Formoterol Combination in Asthma Patients

**DOI:** 10.3390/pharmaceutics12040336

**Published:** 2020-04-09

**Authors:** Hyun-moon Back, Jong Bong Lee, Anhye Kim, Seon-Jong Park, Junyeong Kim, Jung-woo Chae, Seung Soo Sheen, Leonid Kagan, Hae-Sim Park, Young-Min Ye, Hwi-yeol Yun

**Affiliations:** 1Department of Pharmaceutics, Ernest Mario School of Pharmacy, Rutgers, The State University of New Jersey, Piscataway, NJ 08854, USA; hyunmoon.back@rutgers.edu (H.-m.B.); jongbong.lee@rutgers.edu (J.B.L.); lkagan@pharmacy.rutgers.edu (L.K.); 2Center of Excellence in Pharmaceutical Translational Research and Education, Ernest Mario School of Pharmacy, Rutgers, The State University of New Jersey, Piscataway, NJ 08854, USA; 3Department of Clinical Pharmacology and Therapeutics, CHA Bundang Medical Center, CHA University, Seongnam, Gyeonggi-do 13496, Korea; ahkim3478@gmail.com; 4College of Pharmacy, Chungnam National University, Daejeon 34134, Korea; psj7429@cnu.ac.kr (S.-J.P.); jyeongkim@cnu.ac.kr (J.K.); jwchae@cnu.ac.kr (J.-w.C.); 5Department of Pulmonary and Critical Care Medicine, Ajou University School of Medicine, Suwon, Gyeonggi-do 16499, Korea; sssheen@ajou.ac.kr; 6Department of Allergy and Clinical Immunology, Ajou University School of Medicine, Suwon, Gyeonggi-do 16499, Korea; hspark@ajou.ac.kr

**Keywords:** moderate asthma, population analysis, budesonide/formoterol, asthma control test, dose optimization

## Abstract

Exposure-response and clinical outcome (CO) model for inhaled budesonide/formoterol was developed to quantify the relationship among pharmacokinetics (PK), pharmacodynamics (PD) and CO of the drugs and evaluate the covariate effect on model parameters. Sputum eosinophils cationic proteins (ECP) and forced expiratory volume (FEV1) were selected as PD markers and asthma control score was used as a clinical outcome. One- and two-compartment models were used to describe the PK of budesonide and formoterol, respectively. The indirect response model (IDR) was used to describe the PD effect for ECP and FEV1. In addition, the symptomatic effect on the disease progression model for CO was connected with IDR on each PD response. The slope for the effect of ECP and FEV1 to disease progression were estimated as 0.00008 and 0.644, respectively. Total five covariates (ex. ADRB2 genotype etc.) were searched using a stepwise covariate modeling method, however, there was no significant covariate effect. The results from the simulation study were showed that a 1 puff b.i.d. had a comparable effect of asthma control with a 2 puff b.i.d. As a result, the 1 puff b.i.d. of combination drug could be suggested as a standardized dose to minimize the side effects and obtain desired control of disease compared to the 2 puff b.i.d.

## 1. Introduction

Asthma is a chronic condition involving inflammation and hyper-responsiveness of the airway that accompanies symptoms of dyspnea and bronchospasm [1]. Approximately 7.5% of adults are reported to be affected by asthma in the United States, and the global prevalence and incidence of asthma have been steadily rising, with the patients’ burden increasing by around 30% over the past two decades [2,3,4]. It is a complex disease that can be caused by various factors including genetic backgrounds, and has numerous risk factors such as allergens and irritants from the surrounding environment [4]. The rise in prevalence has been related to the recent changes in the lifestyle, which increases exposure to the environmental triggers and it has become a disease that affects almost all populations worldwide [2].

Due to the heterogeneity in the clinical and mechanistic characteristics of asthma, efficient prevention and treatment strategies have proven to be challenging to develop [4,5]. A combination of corticosteroids and β_2_-adrenergic agonists are commonly used to relieve the symptoms of the disease (such as, dyspnea, and bronchospasm) [2,6]. Inhaled corticosteroids can suppress the inflammatory reactions, thereby alleviating levels of eosinophils and other inflammatory factors including eosinophils cationic proteins (ECP), which is one of the major inflammatory biomarkers in asthma [4,6,7,8]. Obstruction of the airways can be relieved and pulmonary function can be increased after inhalation of β_2_-adrenergic agonists, which constrict the smooth muscles of the bronchi. Short-acting β_2_-adrenergic agonists (SABA) are commonly used for a quick relief of acute symptoms, and long-acting β_2_-adrenergic agonists (LABA) are mainly for long-term control of the disease [4,6,9]

During the past decade, the management strategies of asthma have seen a gradual change from alleviating acute symptoms to achieving a more constant control of the disease [10]. According to the Global Initiative for Asthma (GINA) guidelines, the control of the disease can be determined by integrated assessment of the symptoms, lung function, rate of exacerbations, and burden caused in daily life [11,12]. Asthma control test (ACT) was developed and validated as a tool to assess clinical asthma control and clinical outcome (CO) of the disease [10,11]. The ACT has shown reproducibility and reliability, as well as sensitive responses to changes in CO, and therefore has proven its usefulness not only in clinical research but also in general practice for patients with asthma [10,11,13]. With a such quantitative approach in assessing the CO, it would then be of interest to quantitatively evaluate the relationships between the pharmacokinetics (PK), pharmacodynamics (PD), and CO for therapeutic interventions applied for asthma control. In this regard, quantitative models that can describe and predict the PK/PD/CO relationships could be useful in developing therapeutic strategies for asthma control, but such models, especially those for inhaled drugs, are rare in the literature.

Some studies reported that asthma in many patients was not well controlled [14,15], because of its various symptoms in patients. On the other hand, another study reported that a lowered maintaining dose could have the comparable effect of the present dose [16,17], and based on GINA and the British guidelines, the lowest adequate doses of medication need to be used to control asthma [18,19]. Since these drugs have been used long-term to control chronic disease, the minimum effective dose should be considered to avoid local or systemic adverse effects. But those results could not be quantitatively evaluated, because no model can explain the drug exposure-response and clinical outcome relationship. Despite the fact that these drugs are well-known and have been used for a long time, there is a lack of literature for the development of PK and PK/PD models after inhalation for both drugs, because of a lack of a sensitive assay method and ethical issues for the quantitation of drugs at lung in humans. There are previous studies about the population PK model or PK/PD model of each drug. But these studies did not have PD effect data [20,21] or a developed PK/PD model of corticosteroid or LABA separately, even though these drugs are used as a combination therapy [22,23]. And none of them have a clinical outcome data or model, which is the crucial point to determine drug exposure-response relationship with this chronic disease.

In this study, a prospective clinical trial data from adult moderate asthma patients were utilized to establish a population PK/PD/CO model for quantifying the exposure-response and disease progression relationship after inhalation of budesonide (a corticosteroid) and formoterol (a LABA). Following the establishment of the model, simulations were performed to explore the optimized dosing regimen for these fixed dose combination drugs.

## 2. Materials and Methods

### 2.1. Study Design and Subjects

This study was an open-label trial consisting of a 2-week run-in and a 12-week active treatment period (Figure 1). The objective of this clinical study was to analyze the relationship between PK and PD of the inhalation drugs, and to determine which pharmacogenetic type affects the exposure and effect of the drugs. The primary endpoint was the average change of FEV1, depending on the ADRB2 genotype after inhaling budesonide/formoterol for 12 weeks. The secondary endpoints are listed in Appendix A. The study was conducted in accordance with the ethical principles of the Declaration of Helsinki, and the protocol was approved by the institutional review board of Ajou University Hospital (AJIRB-CRO-08-163). Prior to enrolment, written informed consent was obtained from each subject.

Male and female adults (20–70 years of age) with moderately persistent asthma who had been on maintenance therapy for at least 1 month at a constant dosage before the screening were eligible for enrolment. At least 30 days before enrolment and during the 2-week run-in period, patients took a constant inhalation of 800 μg·day^−1^ of budesonide. Subjects who used a SABA for relief of asthma symptoms at least 3 times or have a diurnal variation ≥15% in peak expiratory flow rate (PEFR) over 3 days during the run-in period were enrolled. Additionally, for the analysis of genetic influence, genotypes were screened before enrolment, and the subjects were subsequently grouped by ADRB2 Arg16Gly genotypes: Arg/Arg, Arg/Gly, and Gly/Gly groups. Detailed inclusion and exclusion criteria of study patients are described in Appendix A.

### 2.2. Sampling Time and Analysis

During the 12-week active treatment period, all subjects received two puffs of budesonide/formoterol 160/4.5 μg (Symbicort^®^, AstraZeneca, Sodertalje, Sweden) twice-daily as a maintenance treatment and for relief of symptoms as needed. Since the patients with moderate asthma were recruited as a study subject, this twice-daily dose was selected and sued based on the GINA guidelines and drug label [18,24]. The use of SABA was allowed in case of acute symptoms, and the number of uses was collected. Blood samples were collected to evaluate concentration levels of the drugs and biomarkers at 0 (After run-in period), 10, 20, 30, and 45 min, 1, 2, 3, 4, 5, 6, 7, 9, and 11 h on Days 1 and 7. Additional trough samples were collected on Days 3 to 5 before inhalation. On the day of each visit, subjects inhaled drugs after collecting respective samples.

Plasma concentrations of budesonide and formoterol were determined with previously reported methods [25,26] using validated HPLC-MS/MS (HPLC: Aglient 1200 series, Agilent Technologies, Santa Clara, CA, USA; MS/MS: 4000 Qtrap, Applied Biosystems/MDS SCIEX, Framingham, MA, USA) suited in precision, accuracy, and stability during analytics. The lower limits of quantification of formoterol and budesonide were 1 pg·mL^−1^, and 100 pg·mL^−1^, respectively. To evaluate the lung functions, forced expiratory volume (FEV_1_), maximum mid-expiratory flow (MMEF), and PEFR were assessed at −2, 0, 1, 4, 8, and 12 weeks using calibrated spirometry. Inflammatory biomarkers of sputum eosinophil, sputum ECP, vascular endothelial growth factor, myeloperoxidase, and interleukin-8 were collected at the time of lung function test and quantified following diagnostics standard operating procedure of the hospital. The ACT survey data was collected at 0, 1, 4, 8, and 12 weeks.

### 2.3. Model Development

#### 2.3.1. Structural Model Development

The structural models were sequentially developed from the PK model to PK/PD and PK/PD/CO model, and final PK/PD/CO model parameters were simultaneously estimated. Non-compartmental analysis was done using Day 1 visit data. The PK model of inhaled budesonide and formoterol in patients were firstly developed in the population PK/PD/CO model development process. One-, two-, and three-compartment models were sequentially evaluated as systemic models of each drug. Linear elimination was assumed in the models, as a limited range of dose was evaluated in this study. For absorption of the drugs through inhalation, the lung and gut compartment, which are the major absorption routes for inhalation drug, were added, and a fractionized ratio (FR) of the dose was applied to differentiate the drug absorption through lung (FR) and gastrointestinal tract (1-FR).
Drug dose to lung: FR·DoseDrug dose to gut: (1−FR)·Dose(0<FR<1)

Sputum ECP and FEV1 were selected as the PD markers for budesonide and formoterol, respectively. In order to develop structural PD models, firstly the concentration-time profiles of budesonide and formoterol in the lung compartment were simulated using the developed PK models. These profiles were then used to construct the PD models utilizing the structures of indirect response models [27]. Indirect response models can be either inhibitory or stimulatory: the PD model for budesonide was established to affect the sputum ECP by an inhibition model; and formoterol was modeled to affect the FEV_1_ by a stimulation model. Following establishment of each PD model, all parameters related to the PK/PD models were simultaneously estimated to finalize the models. The PD model for budesonide was described with the following equation:$$\frac{dR}{dt}=k_{in,sputumECP}\times \left(1-\frac{I_{max}\cdot C_{lung}^{BUD}}{IC_{50}+C_{lung}^{BUD}}\right)-k_{out,sputumECP}\times R$$
where, *R* represents the PD response (sputum ECP concentration for budesonide PD response and FEV_1_ for formoterol PD response), *k_in,sputumECP_* and *k_out,sputumECP_* represent production and loss of response rate constants for sputum ECP, respectively; *I_max_* represents maximum inhibitory effect; *IC_50_* represents concentration of budesonide required to exhibit 50% of the maximum inhibitory effect; and $C_{lung}^{BUD}$ represents concentration of budesonide in the lung compartment. Similarly, the PD model for formoterol was described with the following equation:$$\frac{dR}{dt}=k_{in,FEV1}\times \left(1+\frac{E_{max}\cdot C_{lung}^{FMT}}{EC_{50}+C_{lung}^{FMT}}\right)-k_{out,FEV1}\times R$$
where, *k_in,FEV1_* and *k_out,FEV1_* represent production and loss or response rate constants for FEV_1_, respectively; *E_max_* represents maximum stimulatory effect; *EC_50_* represents concentration of formoterol required to exhibit 50% of the maximum stimulatory effect; and $C_{lung}^{FMT}$ represents concentration of formoterol in the lung compartment.

The CO model was added to the developed PK/PD model, and the disease progression model was employed for this purpose. In order to define the rate constant of disease progression in normal status without medication, a literature value was extracted and used as an initial estimate [28]. The PK/PD/CO model was developed so that the PK/PD models for both budesonide and formoterol affect the CO simultaneously. The PK/PD/CO model was described with the following equation (Figure 2):(1)$$\mathrm{CO}\;\left(\mathrm{ACT}\right)=BASE_{ACT}-\alpha \cdot time+\beta_{1}\left(BASE_{sputumECP}-R_{sputumECP}\right)+\beta_{2}\left(R_{FEV1}-BASE_{FEV1}\right)$$
where, BASE*_i_* represents baseline values of *i* (ACT, sputum ECP or FEV_1_) in patients; *α* represents rate constant of natural disease progression; *β* represents offset effect by change of each PD marker; and R*j* represents changed PD value by drug effect for *j* (sputum ECP or FEV_1_).

#### 2.3.2. Covariate Model Development

Effects of the covariates were explored using ‘stepwise covariate modeling method’ by the forward inclusion (*p* < 0.05) and backward exclusion method (*p* < 0.01). The genetic polymorphism of ADRB2 Arg16Gly (Arg/Arg, Arg/Gly, and Gly/Gly) and other demographic information (a total of five; sex, age, weight, height, asthma duration) were applied as the potential covariates for their effects on different parameters.

#### 2.3.3. Statistical Model Development

Following model establishment for the structural model with the population mean, statistical models for the inter-individual variability (IIV), inter-occasion variability and residual variability were developed. Additive, proportional, and exponential error models were explored for IIV and inter-occasion variability of the different model parameters. Additive, proportional, and combined-error models were explored for residual variability. The objective function value was used as a statistical criterion to test and compare each model. If the objective function value was decreased at least 3.84 (*p* < 0.05) for the development for the structural model and covariate forward selection. In addition, when covariate backward elimination was performed to strict covariate selection steps using full model, 6.64 (*p* < 0.01) was used. The final model had the lowest objective function value.

#### 2.3.4. Modeling Methodology and Software

The population PK/PD/CO modeling was conducted using the non-linear mixed effects modeling approach using NONMEM 7.4 (ICON, Hanover, MD, USA) assisted by Perl-speaks NONMEM (PsN) 4.3.0 [29]. Model parameters were estimated by first-order conditional estimation with interaction option. Model diagnostics were performed using the Xpose 4.0 [30]. PsN was used for model evaluation and stepwise covariate modeling method for potential covariates.

### 2.4. Model Evaluation

The visual predictive check (VPC), performed with simulation in 1000 subjects, was plotted for internal evaluation of the model [31] to evaluate the prediction power of the model. Since there was time consuming issues for canonical bootstrap method, non-parametric linearized bootstrap method was also applied for evaluation of the model which was performed 1000 times, and the 95% confidence intervals for the variabilities were obtained [32].

### 2.5. Simulation for Dose Optimization

The finalized population PK/PD/CO model was utilized for simulation to determine the optimal dosing regimen of the budesonide and formoterol combination therapy. The simulation was performed with 1000 subjects in three different dosing regimen groups with 160 µg budesonide and 4.5 µg formoterol: 1) 2 puffs b.i.d. (identical dosing regimen to that used in the clinical trial); 2) 1 puff b.i.d; and 3) 2 puffs q.d. The simulated treatment duration was 12 weeks and ACT was evaluated every 4 weeks. The ACT results were used to predict the disease control, and they were classified as the following: ≥20, well-controlled; 15–19, partially-controlled; <15, uncontrolled [33]. Simulation results from the 2 puffs b.i.d. group were also compared with the observation by the VPC method.

## 3. Results

### 3.1. Clinical Trial Results

The analyses included data from 53 patients with a mean (range) age of 41 years (20–68 years), and baseline FEV_1_ of 85.1% (45.2–111.7%) (Table 1). Plasma concentrations of both budesonide and formoterol reached their peak levels at around 1 h, and decreased rapidly below the limit of quantification within 3.5 h and 2 h, respectively. The FEV_1_ showed steady increase during the active treatment period and was maintained constant after Week 9, but with a large individual variability. Changes of asthma control status and inflammation markers in blood and sputum after treatment were not significant, and also had a large variability. As a result of simple statistical analysis, the patients with ADRB2 Arg16Gly Arg/Arg showed significant positive changes in FEV_1_ (9.6 ± 10.6% (Arg/Arg) vs. 2.2 ± 8.5% (Arg/Gly)) and ACT (2.8 ± 4.6 (Arg/Arg) vs. 0.2 ± 3.5 (Gly/Gly)), compared to those with Arg/Gly or Gly/Gly, respectively. The detailed results of clinical trials were overlapped on VPC plot in Figure 3.

### 3.2. Population PK/PD/CO Analysis

C_max_ and AUC_last_ (Mean ± SE) for budesonide were 1.00 ± 0.27 ng/mL and 3.59 ± 0.35 ng·hr/mL and for formoterol were 4.53 ± 1.43 pg/mL and 20.94 ± 2.27 pg·hr/mL, respectively. The final structure of the population PK/PD/CO model is shown in Figure 2, and estimation results of the model related parameters are shown in Table 2. The plasma compartments for formoterol were best described by a one-compartment model, whereas a two-compartment model was used for budesonide. The inhaled drug was assumed to be absorbed into the plasma compartments via the gut and lung compartments. Fixed values adopted from product monographs were used for the fractionized absorption ratio between lung and gut compartment and their inter-individual variability [34].

Prior to the PK/PD model development, the concentration of each drug in the lung compartment was simulated, because physiologically and mechanistically, it would be the concentration in the lung which determines the PD effects on sputum ECP and FEV_1_ (Figure 2). To simulate the drug concentration in the lung, the amount of both drugs in lung compartment from the final model was divided by the typical volume of the lung (1.1 L) [35]. The concentration-time profile in the lung compartment of budesonide was linked with the PD model for sputum ECP, and that of formoterol with the PD model for FEV_1_. The absorption rate constant of both drugs for the lung compartment (*k_a,Lung_*) were estimated from the final PK model and fixed during PK/PD and PK/PD/CO model development. The IC_50_ value for budesonide on sputum ECP [36] and the EC_50_ value for formoterol on FEV_1_ [37] were fixed using the literature values. Based on the mechanism of each drug and physiological rationale [38,39], the inhibitory effect of budesonide on k_in,ECP_, and the stimulatory effect of formoterol on *k_in,FEV1_* was used for the indirect response model.

In the PK/PD/CO model, both PK/PD models for budesonide and formoterol were structured to affect the ACT scores simultaneously. For the CO model, curative, protective, and symptomatic drug effect models were evaluated, and the symptomatic model was found to provide the best fit with the most appropriate pharmacological rationale. Since there were many parameters in model, we fixed the PK parameters with the parameters obtained from the PK/PD model to get an optimal result and run time. As a result of VPC, it showed that the final population in the PK/PD/CO model was able to reasonably and successfully describe the data observed in the clinical trial (Figure 3).

The effects of genetic polymorphism of the ADRB2 Arg16Gly and other demographic information were evaluated as covariates in the model. Each model parameter was tested separately for its covariate effect, but it was shown to be statistically insignificant for all model parameters. Therefore, covariate effects were not included in the final model.

### 3.3. Simulation Results

A simulation of three different dosing regimens for the budesonide and formoterol combination was performed to predict the most effective dosing regimen for asthma control. One inhalation was assumed to deliver 160 µg budesonide and 4.5 µg formoterol, and the dosing regimens simulated were 2 puffs b.i.d., 1 puff b.i.d, and 2 puffs q.d. ACT scores were evaluated for the simulated outcome of the population PK/PD/CO model. The baseline ACT score at the start of the treatment was assumed to be 19.7 for the simulated patients, as it was the mean baseline value obtained in our clinical trial (Table 1). Simulation results are shown in Figure 4 and the categorized ACT score distribution is shown in Table 3. It was shown that the dose groups with 2 puff q.d. and 1 puff b.i.d. showed comparable effects to the 2 puff b.i.d. administration group, and no significant difference was found in the distribution of patients in the disease-controlled group (Table 3). When the 1 puff b.i.d. and 2 puff q.d. dose groups were compared, 1 puff b.i.d. showed a non-significant difference but slightly a superior effect than 2 puff q.d.

## 4. Discussion

In this study, we have developed a population PK/PD/CO model that describes asthma control achieved by budesonide and formoterol. Inhaled corticosteroids are generally the first-line therapy for asthma and can be combined with a long-acting β_2_ agonist for long-term control of the disease [4]. Since asthma is a chronic disease which cannot be completely cured, investigation of the PK/PD relationships of these drugs can be important to avoid any unwanted side effects, while achieving the control of asthma that is desired [40,41]. The purpose of the inhaled drugs is to exert their pharmacological activities at the target site, which is the lungs, and the complexity of the pulmonary dissolution and absorption processes have hindered development of PK/PD models for inhaled drugs [42]. PK data is often obtained from plasma concentration levels, and identifying the relationship between plasma PK and the PD effects at the lungs can pose additional challenges [42]. In this study, the structure of the PK models for both drugs included lung compartments, which were then used to simulate the concentration-time profiles of drugs in the lungs, which had a more reasonable mechanism for their pharmacological effects. Although it would be challenging to validate the simulated profiles in the lungs, due to the impractically of measuring the concentration levels in human lungs, we have successfully described plasma PK profile in both drugs and from these results of PK model, we have shown that these profiles can be useful in further elucidating the PK/PD/CO relationships of the drugs.

For our PD model, sputum ECP and FEV_1_ were used as the biomarkers representing PD effects of budesonide and formoterol, respectively. FEV_1_ is the most common indicator of the lung function and reduced FEV_1_ suggests obstruction of the airways [4]. Accordingly, there have been previous examples of PK/PD modeling approaches where FEV_1_ was used for the PD effect of bronchodilators [22,23]. In our final model, relationship between formoterol concentration in lung and FEV_1_ was explained by stimulating production of FEV_1_ (*k_in,FEV1_*), rather than inhibiting the loss of FEV_1_. ECP has been reported to be an indicator of airway inflammation, and sputum ECP showed a closer relationship to lung function parameters compared with serum ECP [43,44]. As an inflammatory response marker, sputum ECP has been found in higher levels in asthma patients when compared with normal subjects, and it has also shown to be predicative of asthma control [45,46]. Sputum ECP is a highly variable marker, even within a single subject, and the range was from 2.1 ng/mL to 825 ng/mL in our data, which had around 400 times difference. Because of these limitations from sputum ECP itself and our data, IIV of baseline parameter for sputum ECP (113.9%) and residual variability of the PD model (72.1%) showed a high percentage as well as % RSE value of *k_out,ECP_* (5450%), which meant that the data had an inherently high variability. In addition, since the RSE (%) in NONMEM was calculated by estimated SE (%) divided by estimated parameter value, the small estimated value as absolute magnitude of *k_out,ECP_* (0.00598) was affected for large estimated RSE (%) in our expectation. Since there has been a previous report of correlation between sputum ECP and FEV_1_ [44], a mathematical equation for explaining interaction between the two was applied and compared with the current non-interaction model, but their interaction was statistically insignificant, and therefore was not included in the final model.

While efficacy of a drug can be assessed by different means such as biomarkers, vital signs or CO, the endpoint for a treatment would be to ultimately improve the CO. It is often difficult to quantify the CO and therefore CO has been rarely used in quantitative modeling and simulations [47]. Accordingly, there have been a number of studies which investigated the PK/PD relationship for inhaled drugs, but models that describe the PK/PD/CO relationship are not common [22,23,48,49,50]. In this study, the ACT, a quantitative method of evaluating CO for control of asthma, was utilized, and therefore we were able to establish the PK/PD/CO relationships for budesonide and formoterol. Additionally, the PK/PD/CO model was simultaneously established for the two drugs, which would be necessary since the combination therapy is common for patients with persistent asthma [4]. Since our study did not have a placebo treatment group, placebo data was extracted from another report, which employed a placebo group to explore disease progression [28]. This previous publication report placebo group data for an average of 61 months, which was a long enough period of time to utilize as a base disease progression model. Therefore, we developed a placebo model based on this publication to estimate disease progression parameter (α) and fixed this parameter to develop our PK/PD/CO model. In PK/PD parameters estimated from the final model, formoterol (effect factor on CO: 0.644) was observed to have a larger effect on CO than that of budesonide (0.00008). This result can be explained by the rationale proved in many previous studies, which reported that formoterol combined with budesonide had synergistic effects to contribute to anti-inflammatory effects by increasing the chemokine receptor sensitivity and expression [51,52].

However, the CO model suggested that budesonide might not have a significant disease modifying effect on asthma progression. This finding might be due to the lack of a corticosteroid effect changing the rate of disease progression [53], or high variability among patients. Sputum ECP showed a high variability during the whole study in many patients, and some patients in fact showed a correlation between sputum ECP and ACT scores (the final model assumes inverse correlation). This inconsistent changing of the PD marker may cause reducing effect of sputum ECP on ACT scores. In addition to this, the ACT score as a biomarker that reflects disease progression in this study has been used clinically as a categorical variable to group patients according to severity, but not as a continuous variable proportional to severity. Thus, the ACT score may not be sensitive enough to describe the rate of asthma progression. Another reason might be that, although patients with moderate asthma were enrolled, there may not be significant changes during the observation period in the disease conditions, because the study was conducted in the patients whose disease was at a controlled state after the run-in period. Additionally, the 12-week observation period might not have been a long enough period of time to estimate the rate of disease progression. Asthma is a complex and chronic disease having various sub-phenotypes, and its symptom and progression can be affected by various internal and external factors. The previous studies reported that asthma has an unpredictability with a widely fluctuating nature triggered by heterogeneous external stimuli and the complexity of the respiratory system [54,55,56,57]. Based on recent studies [54,58,59], a further study can be warranted to develop an asthma progression model development to integrate multi-dimensional features of asthma.

Asthma combination therapy can be controversial, in that the combinations might have interaction between the PK of drugs by genetic polymorphism [60,61,62,63] which could affect the PD marker of drugs and eventually the CO as well [64,65,66,67]. During simultaneous quantification of the exposure-response relationship of both drugs, we have evaluated and attempted to quantify the interaction effect: effects of budesonide on FEV_1_; formoterol on ECP; ECP on FEV_1_; and interactive effects of ECP and FEV_1_ to CO. However, we have found that these interactions have non-significant relations, and genetic polymorphism also did not have significant effects in moderate asthma patients. Also, there was no significant effect of covariates on PK, PD, and CO of the drugs, including gender effect.

There were a few limitations of the study that were related to the clinical trial design and the obtained dataset. Firstly, patients were given access to SABA for the alleviation of acute asthmatic symptoms during the trial. Although the action of these agents would be transient, there is possibility that it might have affected the CO. Additionally, since patients were also given budesonide during the run-in period, the baseline of sputum ECP could have been affected. As information on the usage of the short-term relievers was difficult to document, we did not consider it during the model establishment. Secondly, the clinical trial was performed for 20 weeks, including screening periods, and it is considered that the period could have been insufficient to evaluate the curative effects of a chronic disease. In addition, most of the patients enrolled in the trial kept a well-controlled disease state during trial periods, so the CO was mostly consistent throughout the trial. Due to this characteristic of the dataset, which was used for model development, it could be possible that our PK/PD/CO model might not be able to describe a more dynamic change in the CO. Lastly, the clinical trial was conducted within a population with homogenous ethnicity.

The established population PK/PD/CO model was utilized to simulate the predicted CO from different dosing regimens of budesonide and formoterol. It was shown that 1 puff b.i.d. will result in control of asthma comparable to that induced by 2 puffs b.i.d. and q.d. (Figure 3 and Table 3). It was in agreement with a previous report that found that inhaled corticosteroid does not produce a dose-proportionate improvement of asthma [4]. The simulation results of our study also predicted that 1 puff b.i.d. will provide the same benefit as 2 puffs b.i.d., which was the dosing regimen applied in the clinical trial. This could be of importance, as a reduced dosage can potentially minimize any unwanted local or systemic side effects, while maintaining the desired control of the disease.

## 5. Conclusions

The PK of budesonide and formoterol were well described using a two-compartment model and a one-compartment model, respectively. The PD model using biomarkers (sputum ECP for budesonide and FEV_1_ for formoterol) and the CO model using ACT scores were developed using indirect response model and disease progression model, respectively. There was no significant covariate effect, including genetic polymorphism, of ADRB2. As a result of the simulation study, the 1 puff b.i.d. dose showed a comparable effect with the 2 puff q.d. or b.i.d. dose, which meant the 1 puff b.i.d. could be used as a standardized dose for moderate asthma patients to obtain the desired disease control, while minimizing the adverse effect of the drugs. In conclusion, this population PK/PD/CO model can improve the understanding of asthma disease progression, and can be used to predict the optimal dose and subsequent changes of exposure, response, and CO simultaneously.

## Figures and Tables

**Figure 1 pharmaceutics-12-00336-f001:**
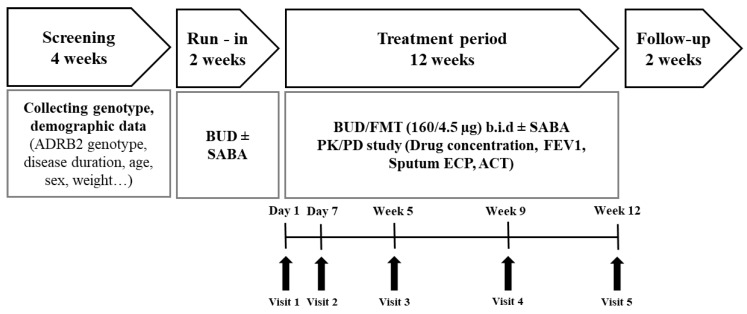
Clinical study design of inhaled budesonide and formoterol in adult moderate asthma patients.

**Figure 2 pharmaceutics-12-00336-f002:**
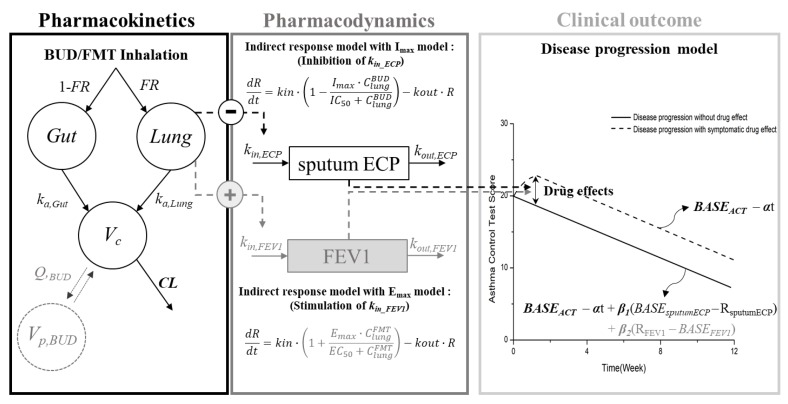
Final model scheme of population pharmacokinetic, pharmacodynamic and clinical outcome model for inhaled budesonide and formoterol in moderate asthma patients.

**Figure 3 pharmaceutics-12-00336-f003:**
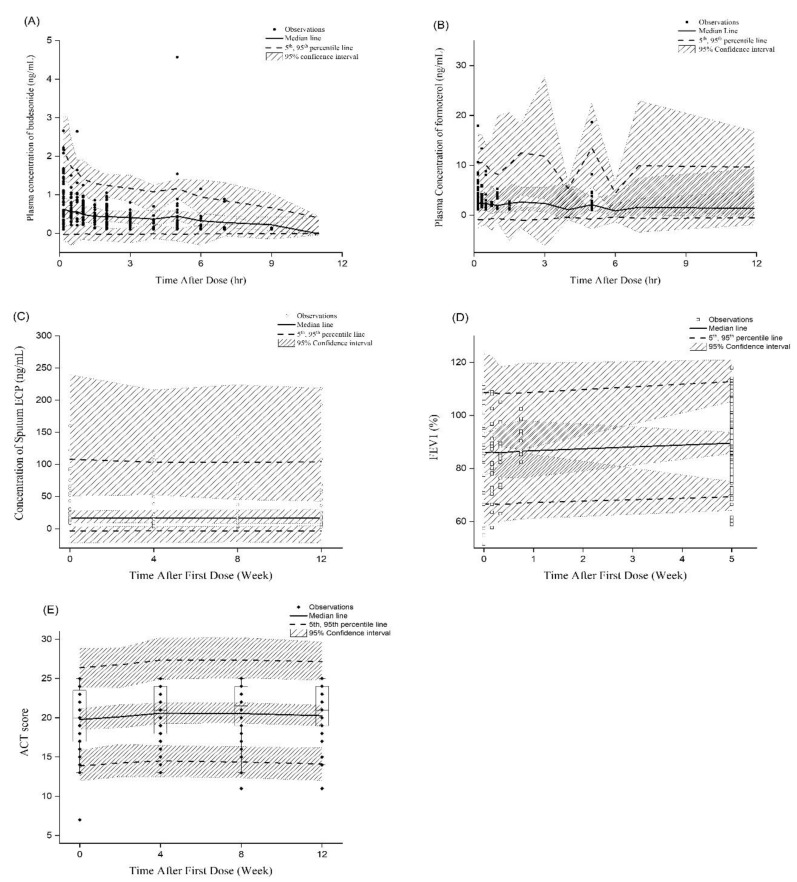
Visual predictive check result of the final model and clinical trial results. (**A**) budesonide PK; (**B**) formoterol PK; (**C**) budesonide PD; (**D**) formoterol PD; (**E**) clinical outcome.

**Figure 4 pharmaceutics-12-00336-f004:**
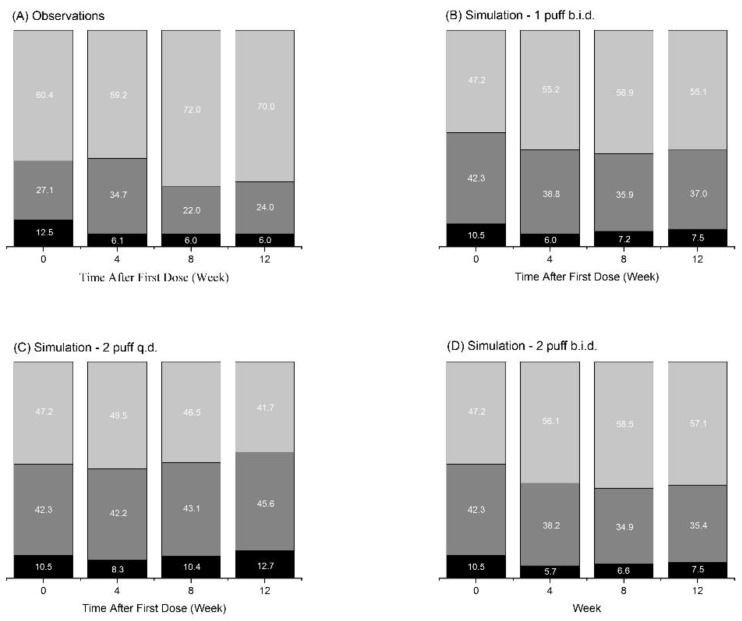
Simulation results plot using the final PK/PD/CO model. (**A**) observation data; (**B**) simulated data by 1 puff b.i.d.; (**C**) simulated data by 2 puff q.d.; (**D**) simulated data by 2 puff b.i.d.

**Table 1 pharmaceutics-12-00336-t001:** Demographic and clinical characteristics.

Variables	All Patients (*n* = 53)
Sex (male/female), n/n (%/%)	33/20 (62%/38%)
Age (years), mean ± SD	41 ± 13
Weight (kg), mean ± SD	70 ± 14
Height (cm), mean ± SD	168 ± 9
ADRB2 genotype (n), AA/AG/GG	16/25/12
Asthma duration (years), mean ± SD	7.0 ± 6.9
Baseline FEV_1_ (%), mean ± SD	85.1 ± 13.8
Baseline sputum eosinophil cationic protein (μg·L^−1^), mean ± SD	51.3 ± 83.2
Baseline Asthma control test score, mean ± SD	19.7 ± 4.1

A: Arg, G: Gly, SD: standard deviation.

**Table 2 pharmaceutics-12-00336-t002:** Estimated parameter from the final PK/PD/CO model for inhaled budesonide and formoterol.

	**Budesonide**					**Formoterol**			
**PK Parameters**	**Value (%RSE)**	**IIV** **(%RSE)**			**Bootstrapped IIV** **(5^th^–95^th^ percentile)**	**Value (%RSE)**	**IIV** **(%RSE)**		**Bootstrapped IIV** **(5^th^–95^th^ percentile)**
*V_c_* (L)	216 ^#^ (20.5%)	93.3% (11.6%)			77.4% (69.6–85.0%)	1250 ^#^ (12.0%)	85.8% (20%)		71.6% (64.7–78.3%)
*CL* (L·hr^−1^)	18.4 ^#^ (29.5%)	-			-	292 ^#^ (16.0%)	91.1% (19.3%)		125.0% (95.3–156.1%)
*k_a,Lung_* (hr^−1^)	19.7 *	-			-	14.8 ^#^ (26.5%)	-		-
*k_a,Gut_* (hr^−1^)	0.00076^#^ (67.5%)	-			-	0.0524 ^#^ (52.0%)	-		-
*FR*	0.38 *	6% *			6% *	0.385 *	-		-
*Q* (L·hr^−1^)	88.3 ^#^ (48%)	-			-	-			-
*V_p,BUD_* (L)	106 ^#^ (37.2%)	-			-	-			-
RV	64.2% (9.0%)				-	80.8% (10.0%)			-
**PD Parameters**	**Sputum ECP**				**Bootstrapped IIV** **(5^th^–95^th^ percentile)**	**%FEV1**			**Bootstrapped IIV** **(5^th^–95^th^ percentile)**
BASE	19.7 ng/mL (38.9%)		113.9% (8.2%)		147.4% (128.9–166.9%)	85.8% (2.0%)	12.9% (10%)		12.7% (11.7–13.7%)
IC_50_ (ng·mL^−1^)	0.025 *		-		-	-	-		
EC_50_ (pg·mL^−1^)	-				-	0.081 *	-		-
k_out_ (hr^−1^)	0.00598 (77.9%)		32.4% (5450%)		26.3% (23.2–30.0)	0.000951 (183%)	110.9% (12.4%)		90.7% (68.1–112.7)
RV	72.1% (5.0%)				-	6.35% (8.0%)			-
**CO Parameters**	**Asthma Control Test**								
	**Value (% RSE)**			**IIV** **(% RSE)**		**Bootstrapped IIV** **(5^th^–95^th^ percentile)**			
BASE_ACT_ (score)	19.70 *			14.9% (9%)		14.8% (13.7–15.8%)			
A	0.00083 *							-	
β_1_	0.00008 (46.9%)							-	
β_2_	0.644 (80.9%)							-	
RV	2.32 (10.7%)					-		-	

$RSE: percentage of relative standard error, RV: residual variability, BAYES: estimated value from Bayesian estimation. ^#^: % RSE of these PK parameters was estimated from the PK/PD model. And these estimated parameters and % RSE were fixed while we developed the PK/PD/CO model. *: Fixed parameter.

**Table 3 pharmaceutics-12-00336-t003:** Simulation results from the final model. Number and percentage of observation were counted from observed data and other scenarios were simulated from the final model (*n* = 1000).

Week	Scenario	Number of Patients			Percentage (%)		
		ACT Score			ACT Score		
		≥20	15–20	<15	≥20	15–20	<15
0	Observation	29	13	6	60.4	27.1	12.5
	1puff BID	472	423	105	47.2	42.3	10.5
	2puff BID	472	423	105	47.2	42.3	10.5
	2puff QD	472	423	105	47.2	42.3	10.5
4	Observation	29	17	3	59.2	34.7	6.1
	1puff BID	552	388	60	55.2	38.8	6.0
	2puff BID	561	382	57	56.1	38.2	5.7
	2puff QD	495	422	83	49.5	42.2	8.3
8	Observation	36	11	3	72.0	22.0	6.0
	1puff BID	569	359	72	56.9	35.9	7.2
	2puff BID	585	349	66	58.5	34.9	6.6
	2puff QD	465	431	104	46.5	43.1	10.4
12	Observation	35	12	3	70.0	24.0	6.0
	1puff BID	551	370	79	55.1	37.0	7.9
	2puff BID	571	354	75	57.1	35.4	7.5
	2puff QD	417	456	127	41.7	45.6	12.7

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
