# Peer review of "Exposure-Response and Clinical Outcome Modeling of Inhaled Budesonide/Formoterol Combination in Asthma Patients"

_pharmaceutics, 2020, doi:10.3390/pharmaceutics12040336_

Round 1

Reviewer 1 Report

Authors Back et al., studied population pharmacokinetics, pharmacodynamics and clinical outcomes of inhaled combined formulation of budesonide and formoterol in asthma patients

The studies are well designed, executed and presented.

However, few available reports and studies on  evaluation of pharmacokinetics of budesonide and formoterol combination using similar modelling approach and clinical studies (Soulele et al., Pulm. Pharmacol. Ther. 2018; 48: 168-178; Chen et al., Clinical Therapeutics 2019; 41 (5): 897: 909), are needed to discuss in the introduction part of this manuscript.

Additionally, I have few minor comments are below:

Page 3. Line 108: LC-MS/MS method used for analysis of pK samples is validated in the lab before application? if yes please provide summary of validation data as supplementary context.

Page 5, Line 194-195: Please deleted the sentence “since there was time consuming…” which is duplicated.

Table 2: Provide additional pK parameters like Cmax and AUC if applicable, as discussed drug concentrations many places.

In discussion part, discus about the gender effect if any on PK/PD of both drugs.

Discuss about the p value obtained and its importance in outcome.

Author Response

Review 1

First of all, we appreciate a lot about reviewer’s enthusiasm. Due to their efforts, our manuscript has been remarkably improved and polished. Hereafter we provide responses one by one for each reviewer’s comments.

[Reviewer’s comment 1]

Authors Back et al., studied population pharmacokinetics, pharmacodynamics and clinical outcomes of inhaled combined formulation of budesonide and formoterol in asthma patients

The studies are well designed, executed and presented.

  • However, few available reports and studies on evaluation of pharmacokinetics of budesonide and formoterol combination using similar modelling approach and clinical studies (Soulele et al., Pulm. Pharmacol. Ther. 2018; 48: 168-178; Chen et al., Clinical Therapeutics 2019; 41 (5): 897: 909), are needed to discuss in the introduction part of this manuscript.

[Answer for reviewer’s comments 1-1]

Thank you for your comment. We’ve added more detailed discussion about previous studies in the introduction part. We’ve added all the recommended references by reviewer as well as some other references about PK/PD model.

Original version

Corrected version

(Red letters were updated from original version)

1. Introduction

None

1. Introduction

There are previous studies about the population PK model or PK/PD model of each drug. But these studies didn’t have PD effect data [20,21] or developed PK/PD model of corticosteroid or LABA separately even these drugs are used as a combination therapy [22,23]. And none of them have a clinical outcome data or model which is the crucial point to determine drug exposure-response relationship with this chronic disease.

Additionally, I have few minor comments are below:

  • Page 3. Line 108: LC-MS/MS method used for analysis of pK samples is validated in the lab before application? if yes please provide summary of validation data as supplementary context.

[Answer for reviewer’s comments 1-2]

Thank you for your comment. Definitely, we performed with validated LC/MS/MS system for both of drugs analysis without any modification of analytical method referenced by 25 and 26. Since validation reports were written by domestic language (in Korean), please understand it not to attached those reports as supplementary context. However, we clarified about the validation in the method section as below:

Original version

Corrected version

(Red letters were updated from original version)

2.2. Sampling time and analysis

Plasma concentrations of budesonide and formoterol were determined with previously reported methods [25,26] using HPLC-MS/MS (HPLC: Aglient 1200 series, Agilent Technologies, Santa Clara CA, USA; MS/MS: 4000 Qtrap, Applied Biosystems / MDS SCIEX, USA) and lower limit of quantification of formoterol and budesonide were 1 pg·ml-1, and 100 pg·ml-1, respectively.

2.2. Sampling time and analysis

Plasma concentrations of budesonide and formoterol were determined with previously reported methods [25,26] using validated HPLC-MS/MS (HPLC: Aglient 1200 series, Agilent Technologies, Santa Clara CA, USA; MS/MS: 4000 Qtrap, Applied Biosystems / MDS SCIEX, USA) suited in precision, accuracy and stability during analytics. The lower limit of quantification of formoterol and budesonide were 1 pg·ml-1, and 100 pg·ml-1, respectively.

  • Page 5, Line 194-195: Please deleted the sentence “since there was time consuming…” which is duplicated.

[Answer for reviewer’s comments 1-3]

Thank you for your comment. We’ve deleted the duplicated sentence

  • Table 2: Provide additional pK parameters like Cmax and AUC if applicable, as discussed drug concentrations many places.

[Answer for reviewer’s comments 1-4]

Thank you for your comment. As you suggested, we’ve added Cmax and AUClast. But since the table 2 is the estimated parameters from the PK/PD/CO model, we’ve added both Cmax and AUClast to the method 2.3.1 and result 3.2 like below.

Original version

Corrected version

(Red letters were updated from original version)

2.3.1. Structural model development

The structural models were sequentially developed from PK model to PK/PD and PK/PD/CO model and final PK/PD/CO model parameters were simultaneously estimated.

3.2. Population PK/PD/CO analysis

None

2.3.1. Structural model development

The structural models were sequentially developed from PK model to PK/PD and PK/PD/CO model and final PK/PD/CO model parameters were simultaneously estimated. Non-compartmental analysis was done using Day 1 visit data.

3.2. Population PK/PD/CO analysis

Cmax and AUClast (Mean SE) for budesonide were 1.00 0.27 ng/mL and 3.59 0.35 ng hr/mL and for formoterol were 4.53 1.43 pg/mL and 20.94 2.27 pg hr/mL, respectively.

5) In discussion part, discus about the gender effect if any on PK/PD of both drugs.

[Answer for reviewer’s comments 1-5]

Thank you for your comment. As a result of covariate analysis, there was no significant covariate effect including gender effect. We’ve added discussion about gender effect like below.

Original version

Corrected version

(Red letters were updated from original version)

4. Discussion

However, we have found that these interactions have non-significant relations and genetic polymorphism also did not have significant effects in moderate asthma patients.

4. Discussion

However, we have found that these interactions have non-significant relations and genetic polymorphism also did not have significant effects in moderate asthma patients. Also, there was no significant effect of covariates on PK, PD, and CO of the drugs including gender effect.

6) Discuss about the p value obtained and its importance in outcome.

[Answer for reviewer’s comments 1-6]

Thank you for your comment. We used objective function value as a statistical criterion while we developed the model. If the objective function value of tested model was not decreased at least 3.84 (p<0.05) that base model, we didn’t include any new parameter or covariate. We’ve added about this model selection criteria to the method section like below.

Original version

Corrected version

(Red letters were updated from original version)

2.3.3. Statistical model development

Objective function value of each model was used as a significance test tool during model development to evaluate and compare each model.

2.3.3. Statistical model development

The objective function value was used as a statistical criterion to test and compare each model. If the objective function value was decreased at least 3.84 (p < 0.05) for development for structural model and covariate forward selection. In addition, when covariate backward elimination was performed to strict covariate selection steps using full model, 6.64 (p<0.01) were used. And the final model had the lowest objective function value.

Reviewer 2 Report

In this manuscript the authors have developed a PK/PD model for Budesonide and Formoterol Combination and linked it to Asthma Control Test Score, a Clinical Outcome. It is a mechanism-based model that links biomarker dynamics to clinical outcome, instead of drug concentrations directly, bringing more insights and physiological-driven quantitative modeling. There are some key details with respect to modeling and overall background and need for the study that the authors have not elaborated. Please find below specific comments:

  • The abstract lacks key details. General terms like exposure-response and disease progression modeling are used. Besides these general terms, the authors need to mention what were the PD markers and Clinical Outcome measures specifically used in the study. In addition, the evaluation of covariates/lack of any covariate effects need to be mentioned. No details on critical model parameters are provided – estimates, CV%, IIV etc. for the reader to assess the model performance. Being a mathematical modeling paper, parameters need to be mentioned in the abstract.
  • While the introduction covers some background on the drug, disease, PD and clinical outcomes measures of interest, the overall rationale for doing the clinical study or the modeling is not clearly stated. On line 78 the authors mention – to optimize dosing regimen – however is an optimization necessary? Is there to optimize the already approved drugs? What were the literature/clinical observations that made them to conclude optimization is necessary?
  • In the introduction, in line 73, the authors mention that such models are rare in literature -without any reference. A reference needs to be added and any model that exists needs to be discussed. Is there no model at all?
  • In the study design section, the clinical study objectives, primary and secondary endpoint details are missing, as are the dose justification details. Why were these doses selected/used?
  • Since these are well known and approved drugs, were there no PK models or prior information available to guide the modeling process? Line 123, the authors mention 1, 2,3 compartment models were tested – was there no literature information available on this that the authors had to test all?
  • Line 182, it is recommended that the authors mention which demographic covariates were evaluated in covariate model development
  • Line 194, 195 – there is repetitive text, please correct the typo.
  • In the results section, line 243, the authors mention that they tested stimulatory effect both on kinFev1 and koutFev1 – these two will yield totally different profiles – do the authors mean inhibition of koutFev1? Also in addition, this decision should also be mechanism and physiologically driven – which process potentially captures the mechanism of the drug better? The authors fail to mention that – it should be discussed, rather than just picking the statistically superior model.
  • In table 2, there are several parameters with superscripts – the footnotes seem to be missing
  • The authors need to address the relatively large %RSE of several parameters such as %RSE on the IIV of kout, it is 5450% - this needs to be discussed.
  • Some parameters have %RSE values listed, while some do not. This needs to be clarified.
  • It would be good to see the PK profile/PK VPC in semi-log scale as well.

Author Response

Review 2

First of all, we appreciate a lot about reviewer’s enthusiasm. Due to their efforts, our manuscript has been remarkably improved and polished. Hereafter we provide responses one by one for each reviewer’s comments.

[Reviewer’s comment 2]

In this manuscript the authors have developed a PK/PD model for Budesonide and Formoterol Combination and linked it to Asthma Control Test Score, a Clinical Outcome. It is a mechanism-based model that links biomarker dynamics to clinical outcome, instead of drug concentrations directly, bringing more insights and physiological-driven quantitative modeling. There are some key details with respect to modeling and overall background and need for the study that the authors have not elaborated. Please find below specific comments:

  • The abstract lacks key details. General terms like exposure-response and disease progression modeling are used. Besides these general terms, the authors need to mention what were the PD markers and Clinical Outcome measures specifically used in the study. In addition, the evaluation of covariates/lack of any covariate effects need to be mentioned. No details on critical model parameters are provided – estimates, CV%, IIV etc. for the reader to assess the model performance. Being a mathematical modeling paper, parameters need to be mentioned in the abstract.

[Answer for reviewer’s comments 2-1]

Thank you for your comment. We’ve updated the abstract as below.

Original version

Corrected version

(Red letters were updated from original version)

Abstract

A population pharmacokinetic (PK), pharmacodynamic (PD) and clinical outcome (CO) model for inhaled budesonide/formoterol was developed simultaneously to quantify the exposure-response and disease progression relationship and evaluate effect of genetic polymorphism in moderate asthma patients. Data were pooled from 53 moderate asthma patients receiving an inhaled budesonide/formoterol combination. A population model was developed, and parameters were estimated using non-linear mixed effect modeling. For evaluating model predictability and parameter stability, visual predictive check and bootstrap were assessed. A one-compartment model for budesonide and two-compartment model for formoterol was used to describe PK of drug and indirect response model and disease progression model were used to describe PD and CO of drug, respectively. Simulation results showed that 1 puff b.i.d. had comparable effect of asthma control with 2 puff b.i.d. This study provides the first population PK/PD/CO model for inhaled budesonide/formoterol combination that can simultaneously describe exposure-response and disease progression in moderate asthma patients. As a result of simulation study from final model, 1 puff b.i.d. of combination drug could be used as a standardized dose to minimize local or systemic side effect and obtain desired control of disease compared to 2 puff b.i.d.

Abstract

Exposure-response and clinical outcome(CO) model for inhaled budesonide/formoterol was developed to quantify the relationship among pharmacokinetics(PK), pharmacodynamics(PD) and CO of the drugs and evaluate the covariate effect on model parameters. Sputum eosinophils cationic proteins(ECP) and forced expiratory volume(FEV1) were selected as PD markers and asthma control score was used as a clinical outcome. One and two-compartment models were used to describe the PK of budesonide and formoterol, respectively. The indirect response model(IDR) was used to describe the PD effect for ECP and FEV1. In addition, the symptomatic effect on the disease progression model for CO was connected with IDR on each PD response. The slope for the effect of ECP and FEV1 to disease progression were estimated as 0.00008 and 0.644, respectively. Total five covariates (ex. ADRB2 genotype etc.) were searched using a stepwise covariate modeling method, however, there was no significant covariate effect. The results from the simulation study were showed that 1 puff b.i.d. had a comparable effect of asthma control with 2 puff b.i.d. As a result, 1 puff b.i.d. of combination drug could be suggested as a standardized dose to minimize the side effects and obtain desired control of disease compared to 2 puff b.i.d.

  • While the introduction covers some background on the drug, disease, PD and clinical outcomes measures of interest, the overall rationale for doing the clinical study or the modeling is not clearly stated. On line 78 the authors mention – to optimize dosing regimen – however is an optimization necessary? Is there to optimize the already approved drugs? What were the literature/clinical observations that made them to conclude optimization is necessary?

[Answer for reviewer’s comments 2-2]

Thank you for your comment. Based on GINA and British guidelines, asthma needs to be controlled with the lowest adequate doses of medication. And there are some reports that lowered dose could have the comparable effect of the fixed dose. Since there is always have a high possibility of occurring system adverse effect since this drug has been used as a long-term medication. Before our studies, there was no study for evaluating exposure-response-clinical outcome relationships, so we’ve used the model not only to explain those relationships but also to optimize the dose based on the simulation. We’ve added about this discussion to the introduction like below.

Original version

Corrected version

(Red letters were updated from original version)

1. Introduction

None

1. Introduction

Some studies reported that asthma in many patients was not well controlled [14,15] because of its various symptoms in patients. On the other hand, another study reported that lowered maintaining dose could have the comparable effect of present dose [16,17] and based on GINA and the British guidelines, the lowest adequate doses of medication need to be used to control asthma [18,19]. Since these drugs have been used long-term to control chronic disease, the minimum effective dose should be considered to avoid local or systemic adverse effects. But those results couldn’t be quantitatively evaluated, because no model can explain drug exposure-response and clinical outcome relationship.

  • In the introduction, in line 73, the authors mention that such models are rare in literature -without any reference. A reference needs to be added and any model that exists needs to be discussed. Is there no model at all?

[Answer for reviewer’s comments 2-3]

Thank you for your comment. We’ve added the reference and discussion about the previous models to the introduction part like below.

Original version

Corrected version

(Red letters were updated from original version)

1. Introduction

None

1. Introduction

There are previous studies about the population PK model or PK/PD model of each drug. But these studies didn’t have PD effect data [20,21] or developed PK/PD model of corticosteroid or LABA separately even these drugs are used as a combination therapy [22,23]. And none of them have a clinical outcome data or model which is the crucial point to determine drug exposure-response relationship with this chronic disease.

  • In the study design section, the clinical study objectives, primary and secondary endpoint details are missing, as are the dose justification details. Why were these doses selected/used?

[Answer for reviewer’s comments 2-4]

Thank you for your comment. We’ve added clinical study objective, primary endpoint and dose justification to the method section and secondary endpoints to the supplementary information.

Original version

Corrected version

(Red letters were updated from original version)

2.1. Study design and subjects

This study was an open-label trial consisting of a 2-week run-in and a 12-week active treatment period (Figure 1).

2.2. Sampling time and analysis

During the 12-week active treatment period, all subjects received two puffs of budesonide/formoterol 160/4.5 μg (Symbicort®, AstraZeneca) twice-daily as maintenance treatment and for relief of symptoms as needed.

Supplementary information 1

None

2.1. Study design and subjects

This study was an open-label trial consisting of a 2-week run-in and a 12-week active treatment period (Figure 1). The objective of this clinical study was to analyze the relationship between PK and PD of the inhalation drugs and to determine which pharmacogenetic type affects the exposure and effect of the drugs. The primary endpoint was the average change of FEV1 depending on the ADRB2 genotype after inhaling budesonide/formoterol 12 weeks. The secondary endpoints were listed in Supplementary Information 1

2.2. Sampling time and analysis

During the 12-week active treatment period, all subjects received two puffs of budesonide/formoterol 160/4.5 μg (Symbicort®, AstraZeneca) twice-daily as maintenance treatment and for relief of symptoms as needed. Since the patients with moderate asthma were recruited as a study subject, this twice-daily dose was selected and sued based on GINA guideline and drug label [18,24].

Supplementary information 1

Secondary endpoints

l  Comparing the improvement of Asthma Control Test score depending on ADRB2 genotype after inhaling budesonide/formeterol 12 weeks.

l  Comparing the observed pharmacokinetic properties of budesonide and formoterol depending on ADRB2 genotype after repeated inhalation of budesonide/formoterol

l  Comparing changed sputum eosinophils cationic proteins depending on ADRB2 genotype

l  Comparing frequency of adverse effects, especially blood potassium and glucose levels and QT interval depending on ADRB2 genotype

  • Since these are well known and approved drugs, were there no PK models or prior information available to guide the modeling process? Line 123, the authors mention 1, 2,3 compartment models were tested – was there no literature information available on this that the authors had to test all?

[Answer for reviewer’s comments 2-5]

Thank you for your comment. Surprisingly, there was no much literature information about PK or PK/PD models especially for the inhalation administration even these drugs were approved around 20 years ago. We’ve found some papers while we developed the model, but since these are the only one or two papers per each drug, we tried to evaluate those 1, 2, 3 compartment model as well. We’ve added this lack of literature to the introduction section like below.

Original version

Corrected version

(Red letters were updated from original version)

1. Introduction

None

1. Introduction

Despite these drugs are well-known and have been used a long time, there is a lack of literature for the development of PK and PK/PD models after inhalation for both drugs, because of a lack of sensitive assay method and ethical issues for quantitation of drugs at lung in human..

  • Line 182, it is recommended that the authors mention which demographic covariates were evaluated in covariate model development

[Answer for reviewer’s comments 2-6]

Thank you for your comment. We’ve added about which demographic covariate were tested like below.

Original version

Corrected version

(Red letters were updated from original version)

2.3.3. Covariate model development

The genetic polymorphism of ADRB2 Arg16Gly (Arg/Arg, Arg/Gly, and Gly/Gly) and other demographic information were applied as the potential covariates for their effects on different parameters.

2.3.3. Covariate model development

The genetic polymorphism of ADRB2 Arg16Gly (Arg/Arg, Arg/Gly, and Gly/Gly) and other demographic information (Total five; sex, age, weight, height, asthma duration) were applied as the potential covariates for their effects on different parameters.

  • Line 194, 195 – there is repetitive text, please correct the typo.

[Answer for reviewer’s comments 2-7]

Thank you for your comment. We’ve deleted the repetitive text.

  • In the results section, line 243, the authors mention that they tested stimulatory effect both on kinFev1 and koutFev1 – these two will yield totally different profiles – do the authors mean inhibition of koutFev1? Also in addition, this decision should also be mechanism and physiologically driven – which process potentially captures the mechanism of the drug better? The authors fail to mention that – it should be discussed, rather than just picking the statistically superior model.

[Answer for reviewer’s comments 2-8]

Thank you for your comment. As you commented, there was a lack of explanation. What we wanted to mention was stimulatory effect of formoterol on Kin,FEV1 were not only physiologically based but also statistically superior. But as you mentioned, the reader could be misunderstood that we didn’t consider or think the physiological rationale or mechanism base. So we’ve deleted and clarified it like below.

Original version

Corrected version

(Red letters were updated from original version)

3.2. Population PK/PD/CO analysis

For the indirect response model for FEV1, we tested stimulatory effect on both kin,FEV1 and kout,FEV1 and the effect on kin,FEV1 described the data better than the effect on kout,FEV1.

3.2. Population PK/PD/CO analysis

Based on the mechanism of each drug and physiological rationale [37,38], the inhibitory effect of budesonide on kin,ECP, and stimulatory effect of formoterol on kin,FEV1 was used for the indirect response model.

9) In table 2, there are several parameters with superscripts – the footnotes seem to be missing

The authors need to address the relatively large %RSE of several parameters such as %RSE on the IIV of kout, it is 5450% - this needs to be discussed.

Some parameters have %RSE values listed, while some do not. This needs to be clarified.

[Answer for reviewer’s comments 2-9]

Thank you for your comment. Regarding high %RSE on IIV of kout,ECP, since the observed data of sputum ECP between study patients had difference more than 400 times from the lower one to the higher one (Range from 2.1 ng/ml to 825 ng/mL), we are thinking that the RSE% of IIV for kout,ECP could have 50 times difference (5450% as a %RSE). In addition, since the RSE(%) in NONMEM was calculated by estimated SE(%) divided by an estimated parameter value, the small estimated value as absolute magnitude of kout,ECP (0.00598) was affected for large estimated RSE(%) in our expectation. We’ve added a more detailed discussion about this to the discussion section.

When we developed the model step by step first, we developed a PK model and then PK/PD model and finally developed PK/PD/CO model and estimated it using the final parameters obtained from PK/PD model. So while we were estimating the parameters for the final model, we fixed PK parameters to get the optimal result and run time since those PK parameters didn’t change that much after we obtained solid parameters from PK and PK/PD model. We’ve updated %RSE values and added footnotes about this process in table 2 and to the result section.

Original version

Corrected version

(Red letters were updated from original version)

3.2. Population PK/PD/CO analysis

None

4. Discussion

Since sputum ECP is a highly variable marker even within a single subject, IIV of baseline parameter for sputum ECP and residual variability of PD model showed high percentage which meant that the data had inherently high variability.

3.2. Population PK/PD/CO analysis

Since there were many parameters in model, we fixed PK parameters with the parameters obtained from PK/PD model to get optimal result and run time.

4. Discussion

Sputum ECP is a highly variable marker even within a single subject and the range was from 2.1 ng/mL to 825 ng/mL in our data which had around 400 times difference. Because of these limitations from sputum ECP itself and our data, IIV of baseline parameter for sputum ECP (113.9%) and residual variability of PD model (72.1%) showed a high percentage as well as %RSE value of kout,ECP (5450%) which meant that the data had inherently high variability. In addition, since the RSE(%) in NONMEM was calculated by estimated SE(%) divided by estimated parameter value, the small estimated value  as absolute magnitude of kout,ECP (0.00598) was affected for large estimated RSE(%) in our expectation

10) It would be good to see the PK profile/PK VPC in semi-log scale as well

Thank you for your comment. As you suggested, we’ve changed PK profile/PK VPC to semi-log scale.
